# Understanding asymmetric synergistic effect between movie actors

**JeongRyun Ko, Jason J. Jung** *

Department of Computer Engineering, Chung-Ang University, Seoul, Korea

* j2jung@gmail.com, j3ung@cau.ac.kr

## Abstract

Synergistic effects between movie actors have been regarded as an important indicator when they are casted in a new movie. People simply assume that synergistic effect is symmetric. The aim of this study is to understand the asymmetric synergy between actors. We propose an asymmetric synergy measurement method for actor's star-power-based costarring movies to understand the synergistic effect. When measuring the synergy, we designed it so that it would be possible to measure the synergy that varies with the time of the costarring movie's release and the synergy of new actors. Measured synergies were analyzed on an actor's synergy and asymmetric synergy between actors to examine the characteristics of highly synergistic actors and the asymmetric synergy between actors. Moreover, we confirmed that measuring synergies asymmetrically demonstrated better prediction performance in various evaluation metrics (accuracy, precision, recall, and F1-score) than measuring synergies symmetrically through the synergy prediction experiment using synergy and asymmetric synergy.

## Introduction

A synergistic effect means a positive influence generated when two objects are together. These synergistic effects occur in various fields, such as science [1–3], mergers and acquisitions [4–6], movies [7, 8], social networks [9, 10], and others fields [11–14], and is used in research. People view the synergistic effect as one that occurs when two objects are together. However, synergistic effects are not the same regarding the degree to which they give and receive from each other, depending on the individual's competence. For example, suppose there are two actors, a veteran actor and a new actor. When a box office (movie success) movie performed by two actors earns higher than the average regarding the star power (the competence of the two actors), it can be determined that the synergy between actors is symmetrically high (Fig 1 (a) and 1(b)). However, we consider the synergy generated by being with a veteran actor from the perspective of the new actor and the synergy generated by being with the new actor from the perspective of the veteran actor. From the perspective of the new actor, the power generated by the veteran actor is high, but from the perspective of the veteran actor, it is difficult to judge that the power generated by the new actor is high. As such, there are cases where the synergy of giving and receiving between two actors is asymmetric (Fig 1(c)). Thus, this study

**Data Availability Statement:** The dataset and source code in this study are available at the following GitHub database (https://github.com/kecau/Interaction-Synergistic-Effect).

**Funding:** Initials of the authors who received award: Jung, J.J. 1) Grant numbers awarded to the

author: NRF-2020R1A2B5B01002207 The full name of each funder: National Research Foundation of Korea 2) Grant numbers awarded to the author: S3228977 The full name of each funder: Ministry of SMEs and Startups The funders had no role in study design, data collection and analysis, decision to publish, or preparation of the manuscript.

**Competing interests:** The authors have declared that no competing interests exist.

focused on the fact that the synergy between the two actors interacts and that there are cases of asymmetry.

To analyze these synergy characteristics, we focused on the synergistic effect between movie actors and various domains. A movie is a work performed by actors, and actors interact with each other while appearing in various movies. If they perform together, synergy is generated. There are previous studies that mention the importance of synergy between actors. These studies have been conducted in connection with box office, which indicate movie success. For example, [7] demonstrated that the synergy of several actors is more important than a single actor when predicting box office success. Additionally, [7] predicted box office success employing the features of the movie. The research result indicated that it is better to proceed with a large number of actors than to predict a box office success with a single actor, which indicates the importance and necessity of studying synergy between actors. Likewise, studies [7, 8] have mentioned the importance of the synergistic effect between actors, but relevant studies on the analysis are difficult to find.

Thus, to analyze the asymmetric synergy between actors, we propose a way to leverage the star power of actors to measure their interaction synergy based on costarring movies. The term star power [15] refers to the actor's competence, and in this study, a filmography is a movie list in which the actor appeared, which is used as star power. A costarring movie is a movie in which two actors appear together. The goals to be achieved by the proposed method are as follows: If the star power is not the same for each actor, the synergy given and received between them should be measured differently. The synergy that new actors without star power receive should be measured. We approach that goal with a method to measure the synergy intensity of each costarring movie by extracting data suitable for five criteria (first, maximum, minimum, average, and frequency) from the star power and a technique to assign new actors with objective star power relevant to the times.

There are conditions to consider when measuring synergy. As in the above example, the synergy between a veteran actor and a new actor should be measured differently, so the actor's competence should also be considered to measure synergy. The actor who gives or receives the best (or worst) synergy to the measured synergy-based actor should be identified differently. Although we have not tried working together between actors, we expect the indirect synergies to be captured through direct synergies. We established the following research questions to verify that the proposed method meets these requirements:

- RQ1. What are the characteristics of highly synergistic actors?

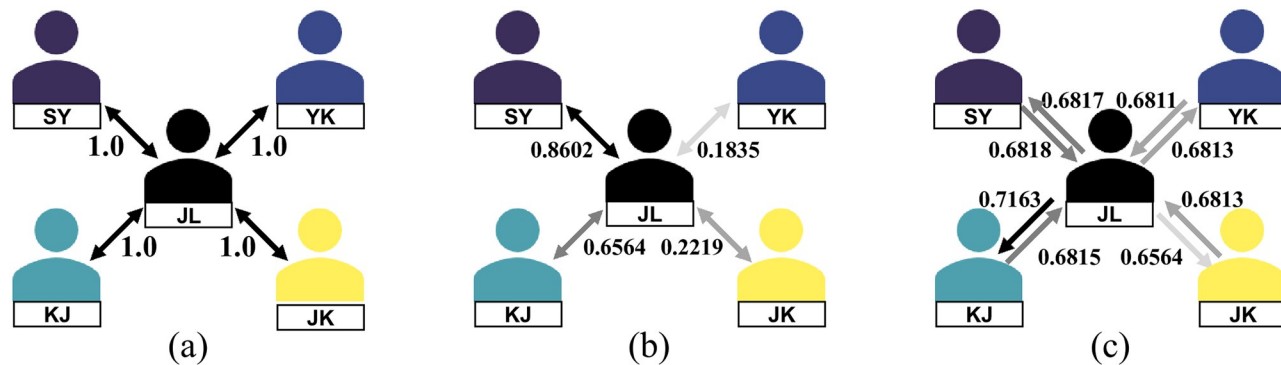

**Fig 1. Different perspectives on synergy.** (a) Symmetry (binary), binary: existence or nonexistence of co-starring movie of the two actors (b) Symmetry (average), average: box office average of co-starring movies of the two actors (c) Asymmetry.

- RQ2. Does an actor that gives good synergy receive good synergy?

- RQ3. Can asymmetric synergies predict synergy between actors well?

We designed the experiment to answer these RQs as follows. First, we analyzed the synergy distribution based on the actor's star power. Through this, it is possible to analyze the relationship between actor characteristics and synergy, and it is possible to ascertain which high and low synergy is densely concentrated on actors with which characteristics. Second, we ascertained the percentage of actors who receive better synergy than they give. The characteristics of the actors in each case were then ascertained. This approach allows us to understand the characteristics of the synergies given and received between actors. Finally, we predicted the synergy between actors. We compared the prediction performance between the proposed method and the baseline, and furthermore, the prediction performance according to the smallest number of costarring movies to determine how many costarring movies are suitable for measuring synergy.

The contributions of this work are as follows:

- We propose a method of measuring synergy that interacts. This method is available for movie actors and various other domains where synergy occurs.

- We promote an understanding of the synergistic effect due to actor characteristics by understanding actor characteristics associated with high synergy and actor characteristics due to the synergy between actors.

- Through synergy prediction experiments, we demonstrate that a method that measures synergy as interaction displays better results than a basic synergy measurement method.

The rest of the paper is organized as follows. Section measurement of synergy describes how to measure the asymmetric synergy between actors. Section evaluation describes the data, analysis, and prediction experiments. Section results and discussion describes and discusses the results of the analysis and prediction. Finally, Section conclusion presents the conclusions of this study and future work.

## Measurement of synergy

In this section, we describe the proposed asymmetric synergy measures. The synergy graph is constructed to analyze the synergy.

### Measurement of asymmetric synergy

The synergy received through the costarring movie must be measured on the actor's star power basis to measure the asymmetric synergy between actors. We measured the synergy intensity received by actors according to five criteria. The costarring movie influence is measured using Eq (1) based on the actor's first movie, maximum box office movie, minimum box office movie, and average star power.

$$\frac{c_b^{A,B} - s_b^A}{c_d^{A,B} - s_d^A} \tag{1}$$

Moreover, Eq (1) measures the synergy intensity actor A receives when appearing with actor B. This measures how much more successful a costarring movie is than the standard actor A's movie, and if success is rapid, the synergy intensity is measured as higher. In addition, $s^A$ is the standard movie in actor A's star power, and $s_d^A$ denotes the release date of that movie.

Finally, we use Eq (2) to measure the frequency of movies that are less successful than the costarring movie at the actor's star power.

$$\frac{|D|}{|f^A|}, D = \{k | k < c_b^{A,B}, k \in f^A\} \tag{2}$$

Additionally, Eq (2) measures the frequency of movies below $c_b^{A,B}$ in terms of actor A's star power. The higher this result is, the more it means that no movie has been more successful than the costarring movie in the past. For the actor, this means that the costarring movie is the best one, and it reveals good synergy through that movie. The final synergy that the actor receives through the costarring movie is obtained by averaging the values for the five criteria.

When measuring synergy, there are cases where the actor is a new actor without star power. If a new actor's star power is set to zero, any new actor who appeared in a best box office movie would be measured as the actor who receives the highest synergy, unifying all actors, and this is not a suitable measurement. If we set the synergy associated with a new actor to zero, this is not a good measurement either because it ignores the synergy generated by that actor's first movie. We grant new actors star power objectively, but the synergy must be measured differently depending on the box office success of the first movie in which they appeared. Therefore, as initial values of star power, we assigned the average release dates and box office values of the previous movies of the costarring movie standard we wanted to measure. This method allowed the star power of new actors to be measured as an objective value according to the period. Thus, the star power values of new actors debuting in the same period are the same. However, the measured synergy about new actors from the best and worst box office movies can be distinguished as measured differently.

## Synergy graph construction

The synergy relationship between actors is represented using a suitable graph structure. The graph is a data structure consisting of nodes and edges that denote the relationships between nodes. Edges can be represented as directed to indicate the directionality between nodes. Depending on the relationship degree between nodes, it can be expressed by a weighted edge. The actor is represented by the node. The interaction–synergy relationship between actors is represented by a directed edge, and the degree of synergistic effect is represented by the weighted edge.

The synergy gained by an actor through another actor was averaged with the synergy value gained per costarring movie and expressed as an edge weight. The range of the averaged synergy is $[-\infty, \infty]$; thus, it was reduced to the $[0, 1]$ range through normalization. Finally, we constructed a synergy graph of the directed weighted graph, revealing the synergy of giving and receiving in the direct edge weight between actor nodes. When analyzing, the synergy between actors was measured only for actors with at least two costarring movies.

## Evaluation

### Using data

This study used data from movies screened in Korea to analyze the synergies between actors. The data were collected using the open API in KOBIS (https://www.kobis.or.kr), which provides information related to the Korean box office. Due to the various data ranges before and after the coronavirus disease 2019 (COVID-19), the data collection period was from 2003 to 2019. The collected data were separated and analyzed with data until 2018, and we predicted

**Table 1. Summary of data.**

|  | The number of movies | The number of actors | The number of synergies |
|---|---|---|---|
| Until 2018 | 3,184 | 2,325 | 24,326 |
| Until 2019 | 3,418 | 3,041 | 32,594 |

2019 data. The information on movies, actors, and synergies for each period is summarized in Table 1.

## Analysis of actor's synergy

To examine the characteristics of highly synergistic actors (RQ1), we analyzed the synergy distribution based on the actor's star power. First, to characterize the actors, we labeled them in two directions according to the number of movies and success score. The number of movies is the count of movies in which the actor has appeared, and the success score is the movie box office average in which the actor appeared. Labeling ensured that the number of actors for each label was as equal as possible.

To analyze the synergy distribution, we employed the node embedding technique, which embeds the actor node of the synergy graph into the latent space according to the synergy relationship. Of the various models for node embedding, we used LINE [16], often referred to as node representation learning [17, 18], which preserves the local and global structure information of a node in a graph.

In addition, LINE embeds the indirect and direct synergy between actors into the actor vector considering the first-order and second-order proximity. Direct synergy refers to the synergy measured between actors, and indirect synergy refers to a predicted synergy between actors who have not appeared in a movie together. The first-order proximity objective function considering the direct synergy is in Eq (3).

$$O_1 = -\sum_{(A,B) \in L} w_{A,B} \log(sigmoid(A, B))$$

(3)

Because Eq (3) is only applicable to the undirected graph, we generated and embedded the undirected graph with direct edge weight averages between actors. The second-order proximity objective function considering the indirect synergy is in Eq (4).

$$O_2 = -\sum_{(A,B) \in L} w_{A,B} \log(softmax(A|B))$$

(4)

In addition, Eq (4) is the objective function that applies the direct synergy between third-party actors to position the indirect synergy information between actors into a latent space. Each proximity was embedded in 128 dimensions, concatenated, and embedded in a 256-dimensional vector for each actor. The parameters set during training for each proximity are as follows: batch size = 512 and epoch = 10. Nodes embedded in the latent space are clustered to cluster actor nodes with similar properties. The cluster character is ascertained by ranking them by the average of the direct synergy between actors in the cluster.

Clustering is one of the unsupervised machine learning techniques. It is a data mining method that finds a cluster by grouping data with similar characteristics in consideration of the characteristics of the data when given data without distinction. The proposed clustering algorithm is various, and there is no algorithm given as the correct answer in all cases, but k-means clustering [19, 20] is usually used. In this study, k-mean++ clustering [21] was used. In

addition, K-means++ improves on the limitations of k-means and is also used in other studies [22, 23]. Before discussing the reason for using k-means++ clustering, k-means clustering is a method of determining a cluster with data close to each center point based on k arbitrary center points. The disadvantage of this method is that it is limited in finding the optimal cluster because it is randomly determined when the center point is initially selected. So, k-means++ clustering came out to compensate for these shortcomings. This method finds k clusters by first randomly selecting one center point from the data, finding data close to the center point, and selecting the farthest data as the next center point. Since the value of k has to be determined, we used elbow method [24] to help determine the number of clusters to classify similar actors.

## Analysis of asymmetric synergy

To determine whether actors who give good synergy receive good synergy (RQ2), we analyzed the asymmetrically interacting synergistic relationship between actors. If interactions represent the synergy between actors, we can determine actors who give the best (worst) synergy to other actors and those who receive the best (worst) synergy. This method was applied to determine the percentage of cases in which the actors received more or less synergy than they gave. To determine the characteristics of each case-specific actor, we ascertained the distribution ratios of actors labeled on a star power basis in the above experiment.

## Prediction of synergy

An experiment was conducted to use data through 2018 to predict the synergy generated in 2019 to examine the predictability of synergy between actors (RQ3). First, it is possible to imagine that the synergy might become more robust as the number of costarring movies rises. Therefore, to determine the optimal minimum number of costarring movies when predicting synergy, we proceeded with the prediction based on the minimum number of costarring movies. We then measured the synergies with optimal minimum number of co-starring movies and proceeded to predict. To assess the degree of predictability, we set the symmetric synergy as the baseline and compared the predictive performance. The first baseline is an undirected unweighted graph measuring the synergy according to the existence or nonexistence of costarring movies with the two actors (Fig 1(a)). The second baseline is an undirected weighted graph measuring the synergy according to the box office average of the costarring movies of the two actors (Fig 1(b)). The proposed method is a directed weighted graph that measures the synergy of interaction based on costarring movies between actors, using the star power of actors (Fig 1(c)).

Predictions were made using a deep neural network (DNN), which uses the input value to predict the target value, a method used for prediction in various studies [25, 26]. The 512 vectors connecting the embedding vector of the two actors were input. The DNN constructed with ReLU activation function between hidden linear layers. The hidden layer comprised five layers: 384, 256, 128, 64, and 32. The loss function used the typical cross-entropy loss for classification, and the optimization was performed using the Adam optimizer. The parameter settings or the optimization were learning rate = 0.001 and weight decay = 1e-7. The batch size was set to 16 and 50 epochs.

To compare the prediction performance of the baseline and proposed methods, we used the accuracy, precision, recall, and F1-score, commonly employed as evaluation metrics [25–28]. Accuracy calculates the percentage of correct predicted results. Precision calculates the probability that the result predicted to be generated is correct. In addition, recall calculates the predicted probability that a generated synergy was generated. As precision and recall have a

**Table 2. Actor's characteristic labeling.**

| The number of movies | | | Score of success | | |
|---|---|---|---|---|---|
| The number of label | Standard | The number of actors | The number of actors | Standard (unit: ten thousand) | The number of actors |
| Many movies | 11 ∼ | 551 | High score | 202 ∼ | 776 |
| Several movies | 5 ∼ 10 | 788 | Middle score | 91 ∼ 202 | 776 |
| few movies | ∼ 4 | 986 | Low score | ∼ 91 | 773 |

trade-off relationship, the F1-score is an index that harmonically averages them into a single numerical value.

## Results and discussion

### Analysis of actor's synergy

Table 2 presents actor labeling in two directions: the number of movies and success score. The labeling was separated into three labels for each direction, classified into the same number of actors as much as possible. The first direction standard is the number of movies, which was categorized as less than four movies (few), less than 10 movies (several), and more than 10 movies (many). The second direction standard is the average movie box office, which was categorized as less than 913,254 (low), less than 2,017,600 (medium), and over 2,017,600 (high).

In addition, Figs 2 and 3 result from counting the actor density by each cluster by label. With this approach, the actor vector generated by embedding the synergy graph was clustered into 12 clusters. The x-axis was arranged from left to right in order of the first to last rank cluster. The cluster rank was ranked by the average of the direct synergy between actors in the cluster. The y-axis indicates counts by actor label. The red dots represent the clusters with the most actors for each actor label.

Further, Fig 2 is the result of counting each actor label according to the number of movies. We confirmed that the many movies label has large numbers in higher synergy clusters than several and few movies labels. This result indicates that actors with numerous movie appearances are likelier to be highly synergetic.

Moreover, Fig 3 is the result of counting each label according to the score of success. We confirmed that the high-score label actors are more frequently in higher synergy clusters than

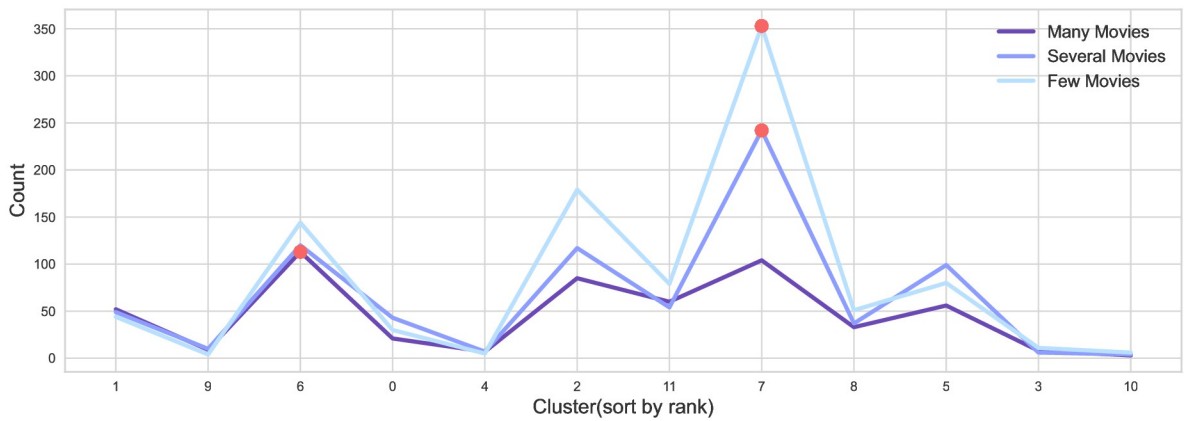

**Fig 2. Cluster distribution by actor label according to the number of movies.**

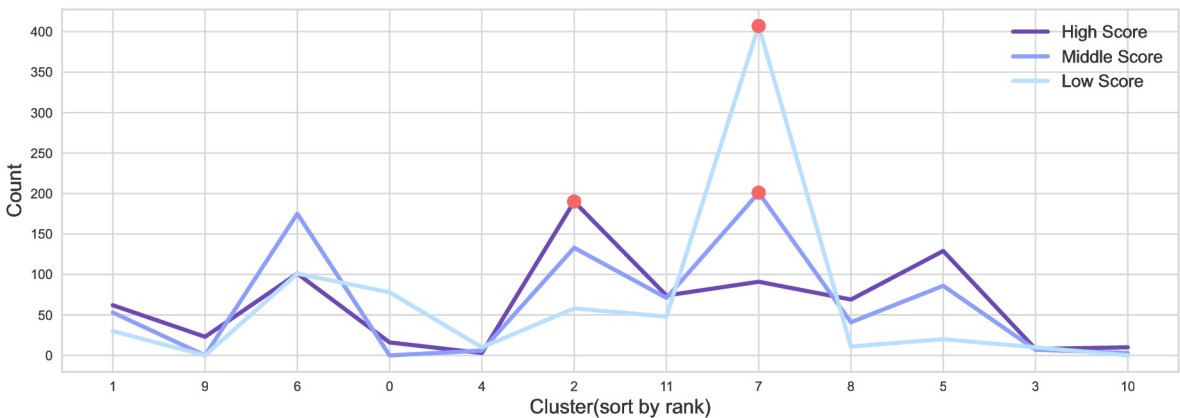

**Fig 3. Cluster distribution by actor label according to the score of success.**

medium- and low-score labels. This result demonstrates that actors who appear in high box office movies are likelier to be high-synergy actors.

## Analysis of asymmetric synergy

Additionally, Fig 4 is the result of analyzing the asymmetric synergy relationship between actors. Referring to the circle chart in Fig 4, the synergy the actor receives is greater (66.1%) than when the synergy the actor gives is greater (33.9%). Through the actor characteristic analysis corresponding to the case when the synergy received by the actor is greater, we confirmed that actors who perform in fewer movies or have lower success scores receive better synergy than they give. In contrast, we confirmed that actors who perform in many movies or have higher success scores give better synergy than they receive.

## Prediction of synergy

In addition, Fig 5(a) is the result of the F1-score of prediction according to the minimum number of movies. Symmetric and asymmetric synergy provide the best synergy prediction

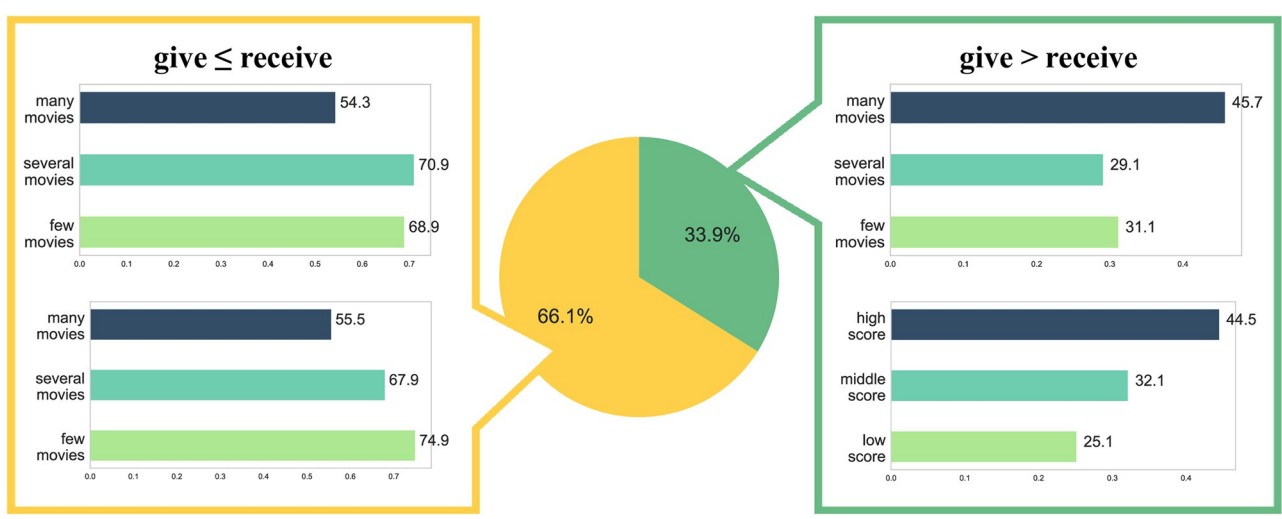

**Fig 4. Analysis of asymmetric synergy.**

performance when the minimum number of movies is two at the effect measurement. It was determined that the optimal minimum number of movies was two by confirming that when three were used, the performance was lower than when one or two was used. Further, Fig 5(b) is the performance evaluation result of the synergy predicted with symmetric synergy (baseline) and asymmetric synergy (proposed), in which the synergy is measured with a minimum

(a)

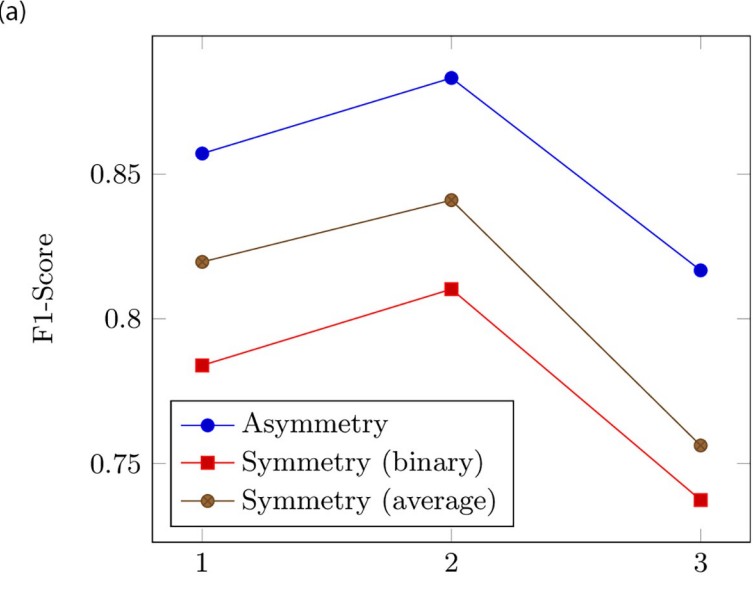

(b)

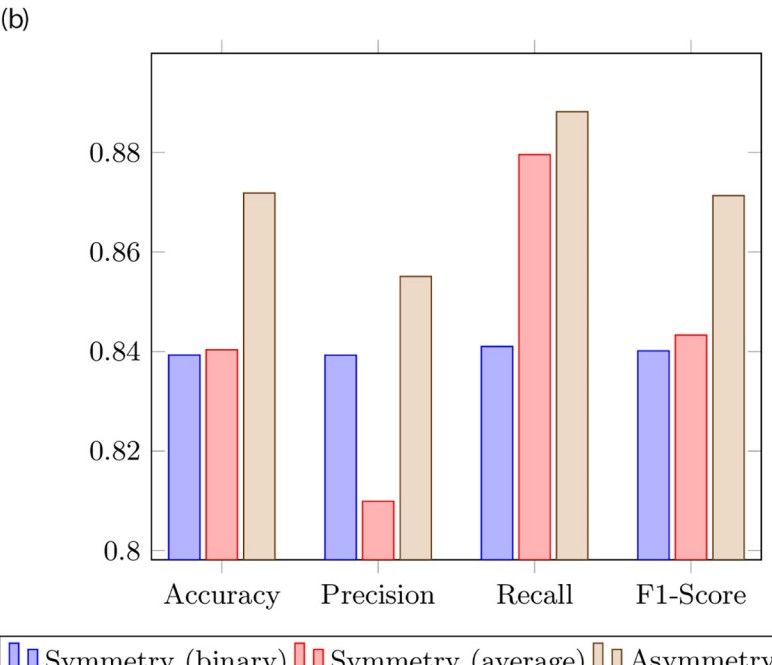

**Fig 5. Result of prediction.** (a) F1-score of prediction according to minimum number of movies (b) Performance comparison of symmetric and asymmetric synergy prediction.

of two movies. Comparing the overall prediction performance confirms that asymmetric synergy has better prediction performance for each evaluation metric.

## Conclusion

This study focused on understanding the synergistic effects between actors. Analyzing the distribution of the actor's star power-based synergy to understand the characteristics of high-synergy actors (RQ1), we confirmed that actors who have appeared in numerous movies and high box office movies are more likely to be high-synergy actors. The results of the asymmetric synergy relationship analysis between the actors to determine whether the actor that gives good synergy receives good synergy (RQ2) indicated that actors giving good synergy were found in about 66% of the cases to be receiving good synergy. In that case, we confirmed that the actors performed in fewer movies and produced lower box office. The results of predicting the synergy between actors (RQ3) confirmed that asymmetric synergy has better prediction performance for all evaluation metrics (accuracy, precision, recall, and F1-score) than symmetric synergy.

A limitation of this study is that a movie is the work of many actors, so there is a difference in the synergy share rate that occurs in the movie. However, objectively evaluating each actor's contribution is difficult; thus, we set the costarring movie as a two actors for this study. Research on measuring synergy contributions within movies is needed to address this limitation.

The method proposed in this study can help recommend actors who give good synergy to the leading actor when the leading actor is determined at the time of movie production. Moreover, the synergy analysis method is not limited to movies but can be applied to various domains in which synergy occurs. Therefore, the plan is to apply the proposed method to synergy-based actor recommendations and various domains in future work.

## Author Contributions

**Conceptualization:** JeongRyun Ko, Jason J. Jung.

**Formal analysis:** Jason J. Jung.

**Funding acquisition:** Jason J. Jung.

**Investigation:** JeongRyun Ko, Jason J. Jung.

**Methodology:** JeongRyun Ko, Jason J. Jung.

**Project administration:** Jason J. Jung.

**Software:** JeongRyun Ko.

**Validation:** JeongRyun Ko.

**Visualization:** JeongRyun Ko.

**Writing – original draft:** JeongRyun Ko.

**Writing – review & editing:** JeongRyun Ko, Jason J. Jung.

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
