## [Decision Letter · Decision Letter 0]

2 Mar 2023

PONE-D-23-03102Understanding asymmetric synergistic effect between movie actorsPLOS ONE

Dear Dr. Jung,

Thank you for submitting your manuscript to PLOS ONE. After careful consideration, we feel that it has merit but does not fully meet PLOS ONE’s publication criteria as it currently stands. Therefore, we invite you to submit a revised version of the manuscript that addresses the points raised during the review process.

We look forward to receiving your revised manuscript.

Kind regards,

O-Joun Lee, Ph.D.

Academic Editor

PLOS ONE

Journal Requirements:

“This study was supported by a grant from the National Research Foundation of Korea (NRF) funded by the Korean government (MSIP) (NRF-2020R1A2B5B01002207).”

“Initials of the authors who received award: Jung, J.J.

Grant numbers awarded to the author: NRF-2020R1A2B5B01002207

The full name of each funder: National Research Foundation of Korea

URL of each funder website: http://www.nrf.re.kr/

Additional Editor Comments:

I would like to ask the authors to revise and improve the manuscript carefully according to the reviewers' comments.

Reviewers' comments:

Reviewer's Responses to Questions

**Comments to the Author**

1. Is the manuscript technically sound, and do the data support the conclusions?

Reviewer #1: Yes

Reviewer #2: No

Reviewer #3: Yes

Reviewer #4: Yes

2. Has the statistical analysis been performed appropriately and rigorously? 

Reviewer #1: Yes

Reviewer #2: Yes

Reviewer #3: Yes

Reviewer #4: Yes

3. Have the authors made all data underlying the findings in their manuscript fully available?

Reviewer #1: Yes

Reviewer #2: Yes

Reviewer #3: Yes

Reviewer #4: No

4. Is the manuscript presented in an intelligible fashion and written in standard English?

Reviewer #1: Yes

Reviewer #2: Yes

Reviewer #3: Yes

Reviewer #4: Yes

5. Review Comments to the Author

Reviewer #1: The manuscript is well-organized and clearly written, with a well-defined research question and hypothesis. The methods are rigorous and appropriate for addressing the research question, and the results are clearly presented and supported by the data. I particularly appreciate the thoroughness of the discussion section, which provides a comprehensive analysis of the findings and their implications.

Overall, I believe that the manuscript is a valuable contribution to the field, and I am confident that it will be well-received by the readership of PLOS ONE. I look forward to seeing it published.

Reviewer #2: The purpose of the study and the results of the experiments are well listed. It seems that the strengths and weaknesses of the thesis have been well considered.

It is considered that it should be submitted after final review.

Reviewer #3: - The manuscript sounds great and the conclusions are drawn properly based on the data.

- The research and study are performed suitably and rigorously.

- The data underlying the methods in their manuscript is fully available.

- The manuscript is presented in standard English.

- Regarding the form, please align the text.

Reviewer #4: The ideas and contributions of the research is presented quite clearly. However, I have some objective assessments to improve the quality of the article as follows:

1. The structure has been clearly divided into sections including an introduction, presentation of the proposed method, experimental results, evaluation, and conclusion. However, related studies have not been introduced in detail and should be described in a separate section immediately after the Introduction.

2. The parameters and variables in the equations should be described in detail which is easy to understand. Authors should write paragraphs containing these equations to increase the coherence of the paper. Besides, describing notations in a table is effective in reading comprehension of formulas.

3. Descriptions for tables and figures should be placed next to this information in the form of captions so that readers can quickly grasp the information.

4. Some passages are written from the author's subjective point of view and have not been mentioned by citations from other articles. Adding citations in the sentences is important to increase the credibility of the article.

5. The author should explain the overview of the major sections, identify the main points in these sections and should pay attention to writing a title that can better cover the content of that section. For example "Evaluation" and "Results and discussion" sections should be explained in detail, the title "Using data" is not really comprehensive, etc.

The above comments are my reviews of this paper, I hope to contribute my efforts to complete high-value research.

6. PLOS authors have the option to publish the peer review history of their article (what does this mean?). If published, this will include your full peer review and any attached files.

Reviewer #1: **Yes: **Dr Wahab Khan

Reviewer #2: No

Reviewer #3: No

Reviewer #4: No

---

## [Author Response · Author response to Decision Letter 0]

6 Mar 2023

Response to Reviewers

We firstly thank all the reviewers for their useful comments. According to the comments, we have carefully revised the paper.

Reviewer #1: The manuscript is well-organized and clearly written, with a well-defined research question and hypothesis. The methods are rigorous and appropriate for addressing the research question, and the results are clearly presented and supported by the data. I particularly appreciate the thoroughness of the discussion section, which provides a comprehensive analysis of the findings and their implications.

Overall, I believe that the manuscript is a valuable contribution to the field, and I am confident that it will be well-received by the readership of PLOS ONE. I look forward to seeing it published.

Reply: Thank you very much for your appreciation.

Reviewer #2: The purpose of the study and the results of the experiments are well listed. It seems that the strengths and weaknesses of the thesis have been well considered.

It is considered that it should be submitted after final review.

Reply: Thank you very much for your positive comments.

Reviewer #3: - The manuscript sounds great and the conclusions are drawn properly based on the data.

- The research and study are performed suitably and rigorously.

- The data underlying the methods in their manuscript is fully available.

- The manuscript is presented in standard English.

- Regarding the form, please align the text.

Reply: Thank you very much for your positive comments.

We have carefully checked the text area in the paper, and align them.

Reviewer #4: The ideas and contributions of the research is presented quite clearly. However, I have some objective assessments to improve the quality of the article as follows:

Reply: Thank you very much for your positive comments.

1. The structure has been clearly divided into sections including an introduction, presentation of the proposed method, experimental results, evaluation, and conclusion. 

Reply: Thank you very much for your positive comments.

However, related studies have not been introduced in detail and should be described in a separate section immediately after the Introduction.

Reply: Thank you for your comments. We have considered this comment very carefully. Different from the other journals, most papers in Plos One has no “Related Work” section. Thus, we have decided to put more related work in “Introduction” section, instead of making a new section for “Related Work”. 

2. The parameters and variables in the equations should be described in detail which is easy to understand. Authors should write paragraphs containing these equations to increase the coherence of the paper. Besides, describing notations in a table is effective in reading comprehension of formulas.

Reply: Thank you for your comments. We have considered this comment very carefully. Since we have only small number of parameters (which can be easily understood), we have decided NOT to put additional description. 

3. Descriptions for tables and figures should be placed next to this information in the form of captions so that readers can quickly grasp the information.

Reply: Thank you for your comments. We have considered this comment very carefully. Since this paper has been working with LaTex, the Figures & Tables are automatically placed. In addition, the editorial work will be supported once the paper is accepted. 

4. Some passages are written from the author's subjective point of view and have not been mentioned by citations from other articles. Adding citations in the sentences is important to increase the credibility of the article.

Reply: Thank you for your comments. We have considered this comment very carefully. In order to improve the credibility of this paper, we have added the following references

- Assenova VA (2018) Modeling the diffusion of complex innovations as a process of opinion formation through social networks. PLoS ONE 13(5): e0196699.

- Pirasteh P et al. (2015) Exploiting Matrix Factorization to Asymmetric User Similarities in Recommendation Systems. Knowledge-Based Systems 83:51-57.

- Grabowicz PA, Ramasco JJ, Moro E, Pujol JM, Eguiluz VM (2012) Social Features of Online Networks: The Strength of Intermediary Ties in Online Social Media. PLoS ONE 7(1): e29358. https://doi.org/10.1371/journal.pone.0029358

5. The author should explain the overview of the major sections, identify the main points in these sections and should pay attention to writing a title that can better cover the content of that section. For example "Evaluation" and "Results and discussion" sections should be explained in detail, the title "Using data" is not really comprehensive, etc.

Reply: Thank you for your comments. We have considered this comment very carefully. The titles of those sections have been corrected. 

The above comments are my reviews of this paper, I hope to contribute my efforts to complete high-value research.

---

## [Decision Letter · Decision Letter 1]

5 Apr 2023

Understanding asymmetric synergistic effect between movie actors

PONE-D-23-03102R1

Dear Dr. Jung,

We’re pleased to inform you that your manuscript has been judged scientifically suitable for publication and will be formally accepted for publication once it meets all outstanding technical requirements.

Kind regards,

O-Joun Lee, Ph.D.

Academic Editor

PLOS ONE

Additional Editor Comments (optional):

I would like to ask the authors to apply the reviewers' minor remarks to the final manuscript.

Reviewers' comments:

Reviewer's Responses to Questions

**Comments to the Author**

1. If the authors have adequately addressed your comments raised in a previous round of review and you feel that this manuscript is now acceptable for publication, you may indicate that here to bypass the “Comments to the Author” section, enter your conflict of interest statement in the “Confidential to Editor” section, and submit your "Accept" recommendation.

Reviewer #1: All comments have been addressed

Reviewer #3: All comments have been addressed

Reviewer #4: (No Response)

2. Is the manuscript technically sound, and do the data support the conclusions?

Reviewer #1: Yes

Reviewer #3: Yes

Reviewer #4: Yes

3. Has the statistical analysis been performed appropriately and rigorously? 

Reviewer #1: Yes

Reviewer #3: Yes

Reviewer #4: Yes

4. Have the authors made all data underlying the findings in their manuscript fully available?

Reviewer #1: Yes

Reviewer #3: Yes

Reviewer #4: Yes

5. Is the manuscript presented in an intelligible fashion and written in standard English?

Reviewer #1: Yes

Reviewer #3: Yes

Reviewer #4: Yes

6. Review Comments to the Author

Reviewer #1: The manuscript has been revised and improved nicely. Authors have considered and explained all the concerns and issues raised by this reviewer. The paper can be accepted for publication.

Reviewer #3: It sounds great.

This article did a good job representing asymmetric synergistic effect between movie actors.

The evaluation is clearly organized.

Reviewer #4: First, I would like to thank the authors who contributed efforts on "Understanding asymmetric synergistic effect between movie actors". This paper present a great idea to understand the asymmetric synergy between actors. However, I have some suggestions to improve the quality of this paper and make it more accessible to readers as follows:

#1. In introduction, from line 36-64, citations should be mentioned more to increase reliability.

#2. Equations should clearly describe their components.

#3. A and B in equation 3 and 4 are not clear in its meaning and how to calculate them. According to LINE model, A and B should be the probabilities of objects A and B. They should be described in detail.

#4. Table captions should include some descriptions about that table.

#5. At the beginning of each section, there should be an overview of that section and subsections to guide the reader.

7. PLOS authors have the option to publish the peer review history of their article (what does this mean?). If published, this will include your full peer review and any attached files.

Reviewer #1: **Yes: **Dr Wahab Khan

Reviewer #3: No

Reviewer #4: No

---

## [Editor Report · Acceptance letter]

11 Apr 2023

PONE-D-23-03102R1 

Understanding asymmetric synergistic effect between movie actors 

Dear Dr. Jung:

I'm pleased to inform you that your manuscript has been deemed suitable for publication in PLOS ONE. Congratulations! Your manuscript is now with our production department. 

Kind regards, 

on behalf of

Prof. O-Joun Lee 

Academic Editor

PLOS ONE